# Long-Term Macrolides in Chronic Respiratory Diseases: Dusk or a New Dawn? A Narrative Review

**DOI:** 10.3390/antibiotics14111061

**Published:** 2025-10-23

**Authors:** Daniele Previtero, Gioele Castelli, Ylenia Padrin, Virginia Santello, Daniele Mengato, Davide Biondini, Graziella Turato, Mariaenrica Tiné, Paolo Spagnolo, Umberto Semenzato

**Affiliations:** 1Respiratory Medicine, Department of Cardiac, Thoracic, Vascular Sciences and Public Health, University of Padua, Via Giustiniani 2, 35128 Padua, PD, Italy; daniele.previtero@aopd.veneto.it (D.P.); gioele.castelli@aopd.veneto.it (G.C.); ylenia.padrin@aopd.veneto.it (Y.P.); virginia.santello@aopd.veneto.it (V.S.); davide.biondini@unipd.it (D.B.); graziella.turato@unipd.it (G.T.); mariaenrica.tine@unipd.it (M.T.); paolo.spagnolo@unipd.it (P.S.); 2Hospital Pharmacy Unit, Azienda Ospedale-Università Padova, Via Giustiniani 2, 35128 Padua, PD, Italy; daniele.mengato@aopd.veneto.it

**Keywords:** respiratory disease, macrolides, azithromycin, bronchiectasis, asthma, COPD, lung transplant, personalized medicine

## Abstract

Macrolides—especially azithromycin—have been increasingly investigated in chronic respiratory diseases for their combined antimicrobial, anti-inflammatory, and immunomodulatory properties. Randomized controlled trials have demonstrated significant reductions in exacerbation frequency in selected patient populations with bronchiectasis, COPD, and asthma, although benefits on lung function and quality of life are variable. Beyond these advantages, concerns remain regarding antimicrobial resistance and uncertainties about long-term safety. Different guidelines across various diseases therefore recommend cautious and selective use of macrolides, with attention to phenotype selection, while alternative or emerging options such as biologics and novel anti-inflammatory agents are reshaping the therapeutic landscape. In this narrative review, we analyze the evidence for macrolide therapy across major chronic respiratory diseases, highlighting both the enduring clinical relevance and the limitations of long-term macrolide therapy, and discussing whether these drugs are approaching their therapeutic dusk or could still open a new dawn for selected patients. To this end we researched EMBASE and PubMed for articles published in English between 2000 and 2020 with restrictions to original articles, and freely between 2021 and 2025.

## 1. Introduction

Since their discovery in the 1950s, macrolides have been used for both antimicrobial and immunomodulatory purposes. The first macrolide, pykromycin—named for its bitter taste (Greek pykro)—was isolated from *Streptomyces* and featured the hallmark macrocyclic lactone ring [1]. This macrolactonic structure, which binds the bacterial ribosomal subunit to inhibit protein synthesis, has been classified based on ring size and the nature of attached amino acids and/or sugars, yielding compounds with distinct pharmacokinetic and pharmacodynamic properties.

The first-generation macrolides—including methymycin, lankamycin, tylosin, carbomycin and the widely used erythromycin—exhibited potent antibacterial activity. However, their clinical utility was limited by poor oral bioavailability and variable kinetics, as they were naturally synthesized by *Streptomyces* spp. These shortcomings spurred the development of semi-synthetic second-generation macrolides—clarithromycin, dirithromycin, roxithromycin, flurithromycin, and azithromycin—which improved on the parent compound’s stability and pharmacological profile. The success of clarithromycin and azithromycin was, however, soon accompanied by rising concerns over antibiotic resistance. This led to the development of ketolides, a third-generation class of macrolides. Among them, telithromycin is the only agent approved to date, displaying enhanced activity against Gram-positive pathogens (including macrolide-resistant *Streptococci*) and selected Gram-negative and intracellular organisms, albeit with notable hepatotoxicity [2].

Beyond their antimicrobial action, macrolides with 14- and 15-membered lactone rings exert substantial immunomodulatory effects. In vitro studies have demonstrated their ability to suppress excessive inflammation by inhibiting the release of pro-inflammatory cytokines from neutrophils and monocytes, and by limiting immune cell recruitment [3]. They also reduce T-cell activation and proliferation, due in part to downregulation of co-stimulatory molecules on antigen-presenting cells and, in the case of azithromycin, reduced expression of major histocompatibility complex class II molecules on dendritic cells [4]. Conversely, macrolides can enhance immune function by promoting phagocytosis, efferocytosis, and microbial clearance. Notably, azithromycin has been shown to restore the impaired phagocytic function of alveolar and monocyte-derived macrophages from patients with Chronic Obstructive Pulmonary Disease (COPD) [5].

Long-term macrolide therapy may also modulate gene expression. Transcriptomic analyses in blood and sputum suggest that azithromycin downregulates key genes involved in antigen presentation, interferon signalling, T-cell responses, and multiple inflammatory pathways [6]. Furthermore, macrolides can reshape the composition of the airway and gut microbiota, potentially restoring the imbalance between commensal and pathogenic bacteria in chronic respiratory disorders [7,8].

On the other side, the activity on microbiota could reduce microbial diversity and alters community composition: the dominance of *Streptococcus* or *Prevotella* species in airway microbiome after 3 months of treatment with low-dose azithromycin has been reported in patients with IPF [9].

Finally, macrolides, in particular azithromycin, are able to reduce *Pseudomonas aeruginosa* quorum-sensing (QS), which is one of its key virulence factors [10]. The precise mechanism of inhibition is still unknown, although it appears to be partly related to interference with ribosomal channel [11]. Recent evidence suggests that the suppression of the QS by macrolides may be transient and strongly influenced by bacterial efflux activity, thus, after macrolide withdrawal, the persistence or upregulation of these efflux systems may facilitate the rapid restoration of QS-regulated virulence traits [12].

Taken together, these preclinical observations suggest that macrolides possess significant immunomodulatory properties, particularly within the airways. These effects—alongside their antimicrobial action—form the rationale for long-term use of macrolides in chronic respiratory diseases. The main antimicrobial, anti-inflammatory, and immunomodulatory mechanisms through which macrolides exert their effects in chronic respiratory diseases are illustrated in Figure 1.

This review explores whether macrolides are approaching their therapeutic dusk, eclipsed by emerging targeted therapies, or whether they may still herald a therapeutic dawn for selected patient populations. We examine current evidence on their use in bronchiectasis, asthma, COPD and other selected respiratory conditions, evaluating their relevance in the evolving therapeutic landscape of respiratory medicine.

## 2. Methods

We performed a narrative review of the literature between January 2000 and December 2020, where filters were applied to exclude case reports, and a second research for articles published between January 2021 and July 2025, with no filter or pre-emptive exclusions, searching PubMed and EMBASE. The search strategy combined the terms “*macrolides*”, “*azithromycin*”, “*clarithromycin*”, “*erythromycin*”, or “*roxithromycin*” with disease-specific keywords including “*bronchiectasis*”, “*asthma*”, “*COPD*”, “*lung transplant*”, “*GVHD*”, and selected rare diffuse lung diseases. During the revision of the literature, only studies conducted in adult populations and in English language were selected. Randomized controlled trials (RCTs), observational studies, systematic reviews, and relevant guidelines were included. In parallel, we extracted studies on alternative or emerging therapies to provide a comparative framework. Data are presented in a narrative and comparative format, with emphasis on clinical outcomes, safety, and efficacy.

## 3. Bronchiectasis

Bronchiectasis is one of the chronic respiratory diseases with the strongest evidence supporting the long-term use of macrolides. Randomized controlled trials have consistently shown that macrolides, particularly azithromycin and erythromycin, significantly reduce the rate of disease exacerbations [14,15,20]. However, the magnitude of benefit varies depending on patient selection, macrolide type, dosing regimen, and treatment duration. Below, we summarize key findings on efficacy, with emphasis on lung function, quality of life, imaging and microbiological bioactivity, and safety. We then discuss the current role of macrolides in bronchiectasis and how emerging therapies might influence their future use.

### 3.1. Effect on Exacerbations

Three landmark trials support macrolide use in bronchiectasis. EMBRACE [14], which used azithromycin 500 mg three times weekly for 6 months, demonstrated a 62% reduction in exacerbations, maintained at 42% 12 months post-treatment. BAT [15] evaluated azithromycin 250 mg daily for 12 months, and showed a 78% reduction in exacerbations, with fewer patients experiencing any exacerbation (46.5% vs. 80%) and a significantly longer time to first event compared with placebo. BLESS [20] assessed the safety and efficacy of erythromycin 400 mg twice daily for 12 months and showed a 43% reduction in exacerbations compared with placebo.

Additionally, smaller studies support the efficacy of macrolides in reducing the rate of exacerbations in patients with bronchiectasis. For example, roxithromycin (150 mg daily) reduced exacerbations by 49% over 12 weeks [21], while an Italian study [16] that used azithromycin 250 mg thrice weekly for 3 months obtained a 92% decrease in exacerbation frequency, although important study limitations included the small sample size and short follow-up. Most RCTs in bronchiectasis have assessed treatment durations ranging from 3 to 12 months. In particular, the EMBRACE trial [14] used azithromycin for 6 months with extended follow-up at 12 months, the BAT trial [15] evaluated azithromycin for 12 months, and the BLESS trial [20] assessed erythromycin for 12 months.

Conversely, in a Japanese study azithromycin provided little benefit in patients who had no history of exacerbations [22], suggesting that long-term macrolides may be less useful in low-risk individuals.

Notably, an individual participant data meta-analysis of macrolide trials confirmed a significant overall benefit of macrolides with a reduction in the rate of exacerbations of approximately 50%. The benefit was consistent across all subgroups, including patients with only 1–2 exacerbations in the previous year, those with *Pseudomonas aeruginosa* infection, who experienced a 63% and 64% reduction, respectively, in the rate of exacerbations [23]. This data supports the use of macrolides also in patients with less frequent exacerbations and in those with *Pseudomonas aeruginosa* colonization, according to the evolving concept of the importance of treating bronchiectasis in earlier phases, like an inflammatory disease.

### 3.2. Effect on Lung Function and Quality of Life

Long-term macrolide therapy has been associated with stabilization or modest improvement of lung function.

In BAT, azithromycin treatment was associated with a gradual increase in FEV_1_ and FVC [15], while BLESS showed a slower decline in FEV_1_ in patients treated with erythromycin [11]. EMBRACE reported improvement in post-bronchodilator FVC but no significant change in FEV_1_ [14]. These effects, though modest, suggest a potential disease-modifying impact of macrolides.

Quality of life, assessed by St. George’s Respiratory Questionnaire (SGRQ), improved significantly in BAT, with 64% of patients achieving the minimal clinically important difference (MCID) (vs. 36% in the placebo group) [15]. Roxithromycin also improved SGRQ scores after 6 months [21]. Erythromycin provided benefit primarily in *Pseudomonas aeruginosa*-positive patients [13]. Reduction in daily sputum volume and purulence, though not always captured in standardized scores, contributes to perceived symptom relief [13,16,17,20,21].

### 3.3. Anti-Inflammatory Effects

Macrolides exert complex effects on the airway microbiome. They reduce pathogen load, particularly *Haemophilus influenzae*, and may achieve temporary eradication of selected organisms including *Pseudomonas aeruginosa* and *Aspergillus* spp. [16,24].

Indeed, rather than outright eradication, macrolides might ‘disarm’ pathogens, explaining why even patients with *Pseudomonas aeruginosa* colonization experienced fewer exacerbations. Quorum-sensing gene expression (e.g., *lasR*, *pqsA*) and biofilm formation are also suppressed, contributing to clinical benefit without bacterial clearance [13].

This mechanism is suggested by the BLESS study, where erythromycin increased *Pseudomonas aeruginosa* detection in patients not previously colonized—possibly due to suppression of other microbes—but paradoxically led to the greatest exacerbation reduction in *Pseudomonas aeruginosa*-infected patients [24].

### 3.4. Effects on Radiological Alterations and Inflammation

Radiological outcomes have not been the primary outcomes of macrolide trials, but some data exists. A post-hoc analysis of BAT showed improvement in total Brody scores after 12 months of azithromycin, driven by reduced parenchymal abnormalities, while mucus plugging and peribronchial thickening improved only marginally [25]. Roxithromycin was also associated with a reduction in bronchial wall thickness and decrease in inflammatory biomarkers (interleukin-8 [IL-8], neutrophil elastase [NE], and matrix metalloproteinase-9 [MMP-9]) [21].

These findings, though preliminary, suggest that macrolides may help stabilize airway damage through anti-inflammatory or anti-remodelling effects.

### 3.5. Safety Profile

Macrolides are generally well tolerated. Gastrointestinal adverse events, such as nausea, epigastric discomfort, and diarrhea, are the most frequent side effects. In EMBRACE [14], these events were significantly more common with azithromycin, but generally mild and transient. In BAT (where patients with liver disease or macrolide intolerance were excluded) [15], patients randomized to azithromycin had an increased risk of diarrhea compared to placebo.

EMBRACE excluded patients with unstable arrhythmias, but in BLESS, which included such patients, QTc intervals remained stable and there was no increased risk of arrhythmias [14,20]. Moreover, a recent observational study from Hong Kong that enrolled over 3000 patients found that long-term macrolide use was associated with a 32% reduction in major adverse cardiovascular events (MACE) without increased risk of ventricular arrhythmias or sudden cardiac death [26]. This study provides additional reassurance on the cardiovascular safety of macrolides.

Resistance remains a major concern. In BLESS, macrolide-resistant *Streptococci* increased from <1% to 28% after 12 months [20], while BAT showed resistance to macrolides increasing to 88% among pathogens from azithromycin-treated patients [15]. Though resistance does not necessarily translate to clinical failure, its implications for public health justify a cautious, targeted use of macrolides. Beyond durations studied in RCTs (12 months), long-term safety and efficacy remain unclear, particularly regarding microbial resistance, microbiome alterations, and other non-infectious risks. Rebound effects, including potentially increased *Pseudomonas aeruginosa* virulence after macrolide withdrawal, possibly related to upregulated quorum sensing pathways, are theoretical concerns.

Overall, the favourable safety profile observed in trials, combined with clinical efficacy, support treatment with macrolides in carefully selected patients, though long-term use requires careful monitoring.

### 3.6. Present and Future of Macrolides in Bronchiectasis

Long-term macrolide therapy remains a cornerstone for patients with bronchiectasis and frequent exacerbations. International guidelines recommend their use in patients with ≥3 exacerbations per year, particularly in the absence of *Pseudomonas aeruginosa* colonization [27,28].

Compared to other available options, macrolides offer several advantages:Greater efficacy than inhaled antibiotics in reducing exacerbations (50–70% vs. ~20–30%) [29]. Inhaled antibiotics achieve only about half the relative reduction observed with macrolides, which explains why guidelines often prefer macrolides as first-line prophylaxis in frequent exacerbators without *Pseudomonas aeruginosa*; however, such differences should be interpreted with caution given the heterogeneity of available trials;Systemic anti-inflammatory effects and immunomodulation, which inhaled antibiotics lack;Low cost and wide availability.

Moreover, bronchiectasis is increasingly recognized as a heterogeneous inflammatory airway disease, and currently macrolides represent the only anti-inflammatory therapy with proven efficacy across different inflammatory endotypes [30], whereas inhaled corticosteroids seem beneficial only in eosinophilic bronchiectasis phenotypes [31]. Recently, a large international observational study combined with a pooled post-hoc analysis of RCTs demonstrated that symptoms, as assessed by the Quality-of-Life Bronchiectasis Respiratory Symptoms Score (QoL-B-RSS), are an independent predictor of future exacerbations, and that patients with a high symptom burden but few prior exacerbations derive similar benefits in terms of exacerbation reduction from long-term macrolide therapy as those with frequent exacerbations [18]. This challenges the current guideline paradigm that restricts macrolide use to frequent exacerbators. Although at the time of writing the European Respiratory Society guidelines on bronchiectasis have not been released yet, it is likely that they will move beyond the current threshold of ≥3 exacerbations per year, recommending long-term macrolides also in patients with fewer exacerbations but a substantial symptom burden, in line with the emerging evidence.

Along with this concept of bronchiectasis as an inflammatory disease, the therapeutic landscape is evolving: brensocatib, a selective oral DPP-1 inhibitor, has shown promise in reducing airway neutrophilic inflammation by inhibiting neutrophil serine protease activation [32]. The Phase 3 ASPEN trial (n = 1721) demonstrated that, compared to placebo, brensocatib (10 or 25 mg daily) reduced exacerbation rates by 20–21%, with a greater proportion of exacerbation-free patients (48.5% vs. 40.3%) and, in patients treated with brensocatib 25 mg, slower decline in FEV_1_ (−24 mL vs. −62 mL per year) [32].

However, the reduction in the exacerbation rate associated with brensocatib use is lower than that observed with macrolides. This can be due to differences in the study populations: the trials of macrolides enrolled patients with more frequent exacerbations, while ASPEN included a broader patient population (≥2 exacerbations per year). Nevertheless, brensocatib appears efficacious in both high- and intermediate-risk patients and may offer an alternative in those with contraindications to macrolides (e.g., NTM infection/suspect, intolerance to macrolides).

In comparing macrolides and brensocatib, several factors emerge:**Efficacy:** Macrolides reduce the rate of exacerbations particularly in high-risk patients. Brensocatib may be more efficacious than macrolides in broader or macrolide-refractory populations.Antimicrobial effects: Macrolides offer both anti-inflammatory and antibacterial activity. Brensocatib does not act on pathogens directly, making it suitable for patients where resistance or infection-related concerns limit macrolide use.Comorbid conditions: Macrolides are beneficial in patients with comorbidities like chronic rhinosinusitis [33], COPD [34] or asthma [35]. At present, the effects of Brensocatib on comorbidities are unexplored as patients with asthma or COPD were excluded from the trials [32]. However, in the phase II study of BI1291583, another DPP-1 inhibitor, patients with a primary diagnosis of asthma or COPD were also included; data from future studies will clarify the effect of these drugs on patients with bronchiectasis and concomitant asthma or COPD [36].Cost and access: Macrolides are widely available and inexpensive. Brensocatib will likely be more costly and restricted to specific indications upon approval. In contrast, macrolides are often prescribed off-label in this setting, a practice that may decrease once an approved therapy for bronchiectasis becomes available.

The potential for combination therapy remains unexplored, although in the ASPEN trial some patients were already receiving background macrolide therapy [32]. Sequential use may represent a pragmatic approach, but clearer guidance is needed to define whether to start with macrolides and escalate to brensocatib in non-responders or vice versa based on patient phenotype and comorbidities.

The advent of targeted anti-inflammatory agents such as brensocatib marks an important milestone in bronchiectasis management. However, long-term macrolide therapy offers a unique combination of antimicrobial and immunomodulatory effects, supported by robust clinical evidence and favorable cost-effectiveness.

Rather than being eclipsed by newer therapies, macrolides are likely to retain a pivotal role in bronchiectasis treatment, particularly in patients with frequent exacerbations and in settings where access to novel therapies remains limited. Their integration with emerging treatments will define the future standard of care, enabling a personalized, phenotype-driven approach.

In conclusion, while the landscape of bronchiectasis therapy is expanding, macrolides remain an affordable, widely available therapy with proven efficacy across inflammatory endotypes. Importantly, their benefits extend beyond frequent exacerbators to highly symptomatic patients with fewer exacerbations, challenging the current treatment paradigm. Far from setting, the therapeutic sun of macrolides continues to rise, and consequently, the use of macrolides in the treatment of bronchiectasis may even increase in the future.

The main clinical scenarios in which long-term macrolide therapy may be considered in bronchiectasis are summarized in Table 1. A comparative overview of macrolides and other present or emerging maintenance therapies in bronchiectasis is reported in Table 2.

## 4. Asthma

Long-term macrolides, particularly azithromycin, have been investigated in asthma for their dual antimicrobial and immunomodulatory effects. Although evidence on exacerbation reduction and symptom control is mixed, they may be considered especially in cases of T2-low asthma. This chapter explores where macrolides currently stand in asthma care, and whether they are approaching therapeutic dusk or poised for a new dawn in selected patient populations.

### 4.1. Effect on Exacerbations

Exacerbation reduction remains a key outcome in asthma. Several trials assessed long-term macrolide therapy with mixed results, with the maximum treatment duration being 48 weeks in the AZISAST trial [37]. Sutherland et al. (2010) [38] and the AZMATICS trial [39] found no significant effect on exacerbations with clarithromycin or azithromycin, respectively. Conversely, AZISAST (2013) showed that azithromycin (250 mg daily for 48 weeks) reduced exacerbations in non-eosinophilic (T2-low) asthma, but not in eosinophilic phenotypes [37].

The AMAZES trial (2017), the largest to date, showed that azithromycin 500 mg three times weekly for 48 weeks reduced exacerbations by approximately 41% versus placebo, with benefit seen in both T2-high and T2-low patients, even if some limitations should be considered, such as the single-centre design and the limited sample size of subgroups [40]. Discrepancy with AZISAST might be explained by differences in phenotype definition (sputum cell counts in AMAZES vs. blood eosinophils in AZISAST) and different treatment schedule [37].

Other studies found no benefit on exacerbation reduction with azithromycin (250 mg twice daily, 3 days/week for 8 months) in severe asthma [41]. A Cochrane review concluded that macrolides probably reduce exacerbations modestly, but evidence remains low-quality due to study heterogeneity [42].

In summary, macrolides, especially azithromycin, can reduce exacerbations in selected patients with severe asthma, but the magnitude of the effect varies according to phenotype and study design.

### 4.2. Effect on Symptoms Control and Quality of Life

Beyond exacerbations, asthma often impairs quality of life (QoL) due to poor symptom control [43]. Macrolides have been evaluated as add-on therapies with variable results. AZISAST reported improvements in symptoms and QoL in the non-eosinophilic subgroup [37], aligning with macrolides anti-inflammatory effects in T2-low asthma [44], while in AMAZES azithromycin improved Asthma Quality of Life Questionnaire (AQLQ) scores compared to placebo, although the change did not reach the MCID [40].

Other trials reported inconsistent findings [39,45]. Trials with clarithromycin yielded mixed results with regard to AQLQ in neutrophilic asthma [38,44]. Roxithromycin showed no significant effect on either daytime or night-time symptoms [46].

Overall, macrolides may modestly improve symptoms and QoL, particularly in T2-low asthma, but effects are inconsistent and phenotype-dependent.

### 4.3. Effect on Lung Function, Airway Hyperresponsiveness and Inflammation

Macrolides have been tested for their impact on lung function and airway hyperresponsiveness (AHR), but data is limited. Most studies, including AZISAST [37] and AMAZES [40], showed no significant changes in FEV_1_ or peak expiratory flow. One study reported modest FEV_1_ and FEV_1_/FVC improvements after 8 months of azithromycin treatment (250 mg twice daily, 3 days/week) [41].

Clearer benefit emerged in infection-driven cases. Kraft et al. [47] showed that clarithromycin significantly improved FEV_1_ only in patients whose BAL was found PCR-positive for *Mycoplasma pneumoniae* or *Chlamydia pneumoniae*, with no benefit in PCR-negative patients, highlighting the potential utility of microbiological testing as treatable trait.

AHR may improve more consistently. Three of four trials assessing AHR showed increased PC_20_ after macrolide therapy [38,44,45,48]. For example, Ekici et al. observed near-doubling of PC_20_ with 8 weeks of low-dose azithromycin. However, benefits may not persist long after discontinuation [49].

Beyond clinical and functional outcomes, some studies have explored the biological effects of macrolides. Azithromycin reduced tumor necrosis factor (TNF) and his receptor (TNFR2) levels in sputum, especially in non-eosinophilic asthmatics (AMAZES [50]). Clarithromycin was also associated with lower inflammatory markers in BAL and sputum, and reduced TNF-α mRNA expression in airway tissue [44,45,47].

Overall, macrolides may attenuate AHR whereas data on lung function improvement is conflicting. Their biological effect on inflammation appears more consistent, although its clinical relevance remains unclear.

### 4.4. Effect on Microbiology and Safety Profile

One possible mechanism for the efficacy of macrolides, particularly in T2-low asthma, is their activity against airway pathogens. Chronic infection with *Mycoplasma pneumoniae* or *Chlamydia pneumoniae* has been linked to neutrophilic inflammation and steroid resistance [51,52]. Kraft et al. [47] showed that clarithromycin improved FEV_1_ only in PCR-positive patients, supporting a microbiologically guided approach. Similarly, Hahn et al. [53] observed symptom improvement with azithromycin in patients with elevated serum anti-*Chlamydia* IgA levels.

Beyond bacteria, macrolides also exert antiviral and immunomodulatory effects, such as enhancement of interferon responses, potentially reducing virus-induced exacerbations [54].

However, their long-term use raises concerns about antimicrobial resistance. The AMAZES sub-study showed that azithromycin reduced *Haemophilus influenzae* load but increased resistance gene expression (erythromycin ribosome methylase [*erm*] and macrolide efflux [*mef*] genes), without reducing total bacterial burden [55]. This has community-level implications, especially with widespread use.

Macrolides are generally safe and well tolerated in asthmatic patients. Across studies, discontinuation rates were similar to placebo. The most common side effect was diarrhea (34% vs. 19% in AMAZES), with rare cases of QT prolongation or hearing loss [40]. Guidelines recommend screening for nontuberculous mycobacteria and baseline ECG before initiating long-term therapy [35].

### 4.5. Current Guidelines and Biologic Therapies: Competition or Combination?

Asthma guidelines and documents reflect the nuanced positioning of macrolides in the era of biologics. GINA 2025 suggests azithromycin (three times weekly for ≥6 months, without specifying dosing) as a step 5 add-on in adults with uncontrolled asthma despite medium or high-dose ICS-LABA, particularly in those with non-T2 inflammation or persistent exacerbations after biologic therapy [35]. In contrast, BTS/SIGN 2024 guidelines do not include macrolides as a recommended option [56]. ERS/ATS 2020 guidelines take an intermediate stance, conditionally recommending macrolides in adults with uncontrolled asthma on maximal therapy [57].

Biologics have transformed the scenario of severe asthma care, particularly in T2-high phenotypes. Agents such as omalizumab, mepolizumab, reslizumab, benralizumab, and dupilumab have shown exacerbation rate reductions up to almost 70% in clinical trials, with greater efficacy in those with higher eosinophil or FeNO levels and many of them also allow for oral corticosteroid tapering [58].

Macrolides have historically been used to fill a therapeutic gap in T2-low asthma, but the advent of tezepelumab—a Thymic Stromal Lymphopoietin (TSLP)-blocking biologic—has shifted this landscape. In the NAVIGATOR trial, tezepelumab reduced exacerbations by 56%, independently of eosinophil counts or allergic status [59]. Pooled analyses confirmed efficacy across T2-high and T2-low subgroups [60].

This raises a key question: if tezepelumab effectively targets T2-low disease, what is the remaining role for macrolides? In real-world practice, patient-specific factors such as cost, accessibility, and overlapping endotypes still matter. Azithromycin, being inexpensive and widely available, remains an attractive option. Even if not equivalent to biologics, it could be considered when they are inaccessible or insufficient.

Moreover, biologics and macrolides are not mutually exclusive. A study by Lavoie et al. [61] showed that adding azithromycin to biologic therapy in patients with persistent exacerbations led to further clinical improvement, especially in those with low FeNO or positive sputum cultures, suggesting they can complement each other. Interestingly, all patients in the study had T2-high asthma, suggesting a potential role for macrolides also in this disease phenotype.

These findings must now be reconsidered in the light of tezepelumab, which has demonstrated efficacy in both T2-high and T2-low patients. A critical knowledge gap remains: when should physicians consider adding azithromycin in a patient already on biologic instead of switching to a biologic that also targets non-T2 inflammation (e.g., tezepelumab)? Further studies are needed to guide this decision-making process.

### 4.6. Future Perspectives: Macrolides at Sunset or a New Dawn in Asthma?

Considering the above evidence, macrolides occupy a unique, somewhat transitional place in asthma therapy. Their efficacy is now eclipsed by biologics that can achieve higher exacerbation reduction rate [58]. Biologics also have more profound effects on lung function and can act as steroid-sparing [58].

In a post-hoc analysis of AMAZES, about 50% of azithromycin-treated patients achieved “clinical remission” (well-controlled disease with no exacerbations for 6 months) versus 39% in the placebo group [62], suggesting that remission is an attainable goal with macrolide therapy. However, modern biologics are also aiming for remission: recent data suggest that roughly one in five severe asthma patients achieve a state of clinical remission on biologic therapy [63]. Thus, macrolides and biologics might both be avenues to the ambitious goal of asthma remission, but biologics have the momentum of precision targeting behind them.

So, are macrolides approaching their therapeutic sunset? Not entirely. While it is true that widespread use of macrolides may decline as more patients are placed on biologics (especially with the advent of tezepelumab covering the T2-low treatment gap), macrolides are likely to remain relevant in specific scenarios. They include:**T2-low or Mixed-phenotype Asthma:** Patients with neutrophil-predominant asthma, chronic bronchitis symptoms, or evidence of airway infection may continue to benefit from macrolides.**Resource-Limited Settings:** In many parts of the world, access to biologics is limited by cost or infrastructure.**Adjunct to Biologics:** As illustrated by Lavoie et al., macrolides might play a role as adjunctive therapy in patients who only partially respond to biologics [61].

In conclusion, macrolides in asthma are neither a panacea nor obsolete. Their therapeutic sun is not setting just yet, but it rises only for the right patient at the right time. As novel biologics and precision drugs expand, the role of macrolides will likely narrow to specific niches where their unique benefits outweigh the risks. In the balance of “dusk or new dawn,” macrolides may represent a persistent twilight: a lasting glow in the asthma armamentarium, even as the brighter lights of targeted therapies illuminate the path forward.

The main clinical scenarios in which long-term macrolide therapy may be considered in asthma are summarized in Table 3. A comparative overview of macrolides and biologic therapies available for asthma is presented in Table 4.

## 5. Chronic Obstructive Pulmonary Disease

Chronic Obstructive Pulmonary Disease (COPD) is characterized by chronic symptoms and acute exacerbations that worsen clinical outcomes and increase mortality [34]. Preventing exacerbations is a major therapeutic goal. This chapter summarizes current evidence—including exacerbation reduction, clinical outcomes, microbiological effects, and safety—and compares macrolides with newer treatments.

### 5.1. Effect on Exacerbations

Several studies support long-term macrolide therapy in reducing COPD exacerbations. Early studies with erythromycin (250 mg twice daily for 1 year) showed a 35% reduction in moderate-to-severe exacerbations and hospitalizations [65]. Similarly, low-dose erythromycin (200–400 mg daily) reduced exacerbations compared to placebo in a study of 109 patients (11% vs. 56% with ≥1 event) [66], possibly by disrupting the infection–exacerbation cycle, which is often triggered by viral Upper Respiratory Tract Infections [67].

Clarithromycin has been less investigated. A small RCT (n = 67) testing 3-month extended-release clarithromycin found no significant reduction in exacerbations [68], probably because of the short duration and limited statistical power of the study.

Azithromycin has been extensively studied in COPD. In the landmark trial by Albert et al. in 1142 patients, daily azithromycin (250 mg daily for 1 year, representing the RCT with longest duration in COPD) reduced exacerbations by 27% and extended median time to first event (266 vs. 174 days) [69], with benefit observed across several subgroups except current smokers [69,70]. The COLUMBUS trial (500 mg three times weekly) showed a 40% reduction in the exacerbation rate and longer time to first event; 28% of patients remained exacerbation-free vs. 7% in placebo over the 12-month treatment period [71]. Follow-up data suggested no further gain with prolonged treatment beyond 1 year [72].

In patients with acute exacerbations, the BACE trial tested azithromycin during and after hospitalization: 500 mg daily ×3 days, then 250 mg every other day for 3 months. Azithromycin reduced treatment failure (49% vs. 60%), mainly by means of fewer readmissions and treatment intensification [73]. Benefits disappeared after stopping treatment, as time-to-event curves converged at 6 months [73]. Post-hoc analysis found greater benefit in patients with high blood levels of C-reactive protein or low blood eosinophils count (BEC), hinting at biomarker-driven responses [74].

Real-world studies in GOLD E patients on triple therapy confirm sustained benefit, with reductions in exacerbations (66–70%) and hospitalizations, though partial waning of the beneficial effect occurs after 24–36 months [75,76].

Phenotype may influence response. A post-hoc analysis of BACE showed greater benefit in patients with low BEC [74], while in COLUMBUS, those with ≥2% eosinophils and milder COPD benefited more in terms of reduction in exacerbations [77]. These differences may reflect study designs and populations, but overall, macrolides remain efficacious in high-risk patients regardless of eosinophil count. Even in tracheostomized patients, azithromycin reduced exacerbations and healthcare use over 6 months [78].

In summary, long-term macrolides, particularly azithromycin, consistently reduce the frequency of COPD exacerbation.

### 5.2. Effect on Lung Function, Symptoms, and Inflammation

Macrolides have modest impact on lung function. Trials with erythromycin and azithromycin showed no significant changes in FEV_1_ over time [65,71,79].

Effects on symptoms and quality of life are inconsistent. Most studies, including BACE and Banerjee et al., reported no meaningful differences in CAT, mMRC, EQ-5D, or SGRQ scores, aside from small improvements in SGRQ “symptoms” domain or cough-specific QoL in selected cohorts [68,71,73,78,80]. Symptom relief, when present, is modest and variable.

In contrast, macrolides consistently reduce airway and systemic inflammation. Erythromycin decreased sputum neutrophils and IL-17/IL-23 levels, with rebound after discontinuation [79]. Azithromycin lowered serum IL-6, IL-13, and TNFR2 in severe COPD, with partial effects on sputum biomarkers [75]. These findings confirm anti-inflammatory activity despite limited symptomatic improvement.

In summary, while macrolides do not significantly improve airflow limitation or daily symptoms, while the effect on modulating airway inflammation, supports their use primarily for reducing exacerbations.

### 5.3. Effect on Microbiology and Antibiotic Resistance

A key concern with long-term macrolide use is the risk of selecting resistant organisms and altering the airway microbiome. Despite low doses, does antimicrobial pressure persist?

Short-term data suggests limited microbiological impact. A 3-month trial of clarithromycin showed no reduction in sputum bacterial load or shift in pathogen spectrum, and no new macrolide-resistant Gram-negatives [68].

Longer studies show mixed results. In COLUMBUS, fewer azithromycin-treated patients carried macrolide-resistant bacteria than placebo, possibly due to reduced pathogen load [71]. In contrast, Albert et al. found increased nasopharyngeal carriage of resistant organisms [69], with other studies reporting increased resistance genes (*erm*, *mef*) post-treatment [81].

Macrolides may shift microbial ecology. Azithromycin decreased isolation of *Haemophilus influenzae* and *Moraxella catarrhalis*, but increased that of *Stenotrophomonas maltophilia* and other resistant Gram-negatives [75]. Pomares et al. observed a reduction in *Haemophilus*- and *Moraxella*-related exacerbations, but more Gram-negative infections over time [76]. Data on *Pseudomonas aeruginosa* are conflicting, with one study showing an increase [76] and another showing stability [75].

Genetic analyses confirm adaptation: *Haemophilus influenzae* strains developed 23S rRNA mutations and other resistance-related changes during azithromycin therapy, even without overt resistance in cultures [82].

In BACE, no significant increase in resistance emerged over 3 months, though baseline *Haemophilus influenzae* colonization was higher in the azithromycin group [73]. A 6-month study in tracheostomized patients showed no major microbiological shifts, suggesting short courses may carry lower resistance risk.

In summary, prolonged macrolide therapy can select resistant strains and reshape airway microbiota, though risk varies by duration and population.

### 5.4. Safety Profile

Low-dose macrolides are generally well tolerated in COPD. Controlled trials report no increase in serious adverse events. Seemungal et al. found no liver or cardiac toxicity with erythromycin, and no discontinuations due to side effects [65]. Similarly, the BACE trial found no excess adverse events or QT prolongation with azithromycin over 3 months [73].

On the other hand, in COLUMBUS gastrointestinal (GI) issues were more frequent, though rarely severe, in patients treated with azithromycin. Similarly, in the study by Pomares et al. [76], no serious adverse event were observed, but GI issues and hearing loss occurred in both the long-term and short-term cohort, at frequencies < 10%; one patients had an increase in liver enzymes, with need for reduction in drug dose without discontinuation. Albert et al. [69] observed a mild increase in the rate of ototoxicity in the treatment group compared to placebo (25% vs. 20%), but it was mostly reversible after treatment cessation.

No major arrhythmias were seen in COPD trials, likely due to exclusion of high-risk patients [65,73]. Caution remains warranted in individuals with cardiac comorbidities or QT-prolonging medications.

In conclusion, macrolides are well tolerated in COPD. Yet, monitoring for hearing and cardiac effects remains advisable during long-term use. This aspect should be underscored, considering that patients with COPD are generally elderly and affected by comorbidities (particularly cardiovascular ones), often undergoing polypharmacy and commonly experiencing age-related physiological hearing loss.

### 5.5. Emerging Therapies

The expanding landscape of anti-inflammatory therapies in COPD is redefining the role of macrolides. Several newer agents aim to reduce exacerbations without the downsides of chronic antibiotic use. Below, we compare these therapies to macrolides in terms of efficacy and target populations.

Oral PDE Inhibitors: Roflumilast, an oral PDE4 inhibitor, reduces exacerbations by 14–26%, especially in patients with chronic bronchitis (CB) and/or ICS use [83,84]. However, GI side effects and weight loss frequently lead to discontinuation, especially in real-world settings [85]. A retrospective study found higher hospitalization and mortality with roflumilast vs. azithromycin [86]; the ongoing Phase III RELIANCE trial (NCT04069312) has been specifically designed to compare the efficacy of roflumilast vs. azithromycin in preventing hospitalization or death in patients with high-risk COPD with a previous hospitalization for a COPD exacerbation [87]. Due to its modest efficacy and tolerability concerns, roflumilast is often reserved for patients in whom macrolides are contraindicated.

Biologic Therapies (anti–IL-4Rα and IL-5 Inhibitors): Although eosinophil levels in COPD are not generally elevated compared to the general population, a subset of patients displays eosinophilic inflammation and features of type 2 (T2) inflammation [88,89]. The relevance of blood eosinophils as a biomarker in COPD remains debated, partly due to the conflicting data for correlation of blood and airway eosinophils [88,90]. Nonetheless, T2 biologics have been studied in COPD.

Mepolizumab (anti–IL-5) showed mixed results in early trials (METREX, METREO) but reduced exacerbations by 21% in patients with blood eosinophils ≥300 cells/µL (with or without CB) in the recent MATINEE trial [91,92,93]. Dupilumab (anti–IL-4Rα) showed 30–34% reduction in exacerbations, improved FEV_1_, and QoL in eosinophilic, CB-phenotype COPD patients in the BOREAS and NOTUS trials [94,95].

While the magnitude of exacerbation reduction observed with dupilumab appears comparable to that of macrolides, the efficacy of macrolides in COPD with T2 inflammation remains controversial, particularly when stratified by eosinophil levels, as highlighted in this review. In patients with evidence of T2 inflammation, there is robust rationale for using targeted biologics, particularly dupilumab, whose mechanisms extend beyond eosinophil depletion and include inhibition of key cytokine pathways involved in mucus production, airway remodelling and inflammation (reviewed in [87]). To date, no studies have assessed the combination of macrolides and biologics in COPD, leaving important clinical questions unanswered. Therefore, treatment decisions should be guided by individual patient characteristics, including inflammatory profile and comorbid conditions as well as cost considerations and accessibility.

Inhaled PDE Inhibitors: Ensifentrine (PDE3/4 inhibitor) showed lung function improvement and 40% exacerbation reduction in ENHANCE-1/2 [96,97], though many patients were not on full inhaled therapy, limiting generalizability of findings. Tanimilast, a selective PDE4 inhibitor, had modest overall effects but large exacerbations reduction (49–73%) in patients with CB and BEC ≥ 150 cells/μL [98]. Both agents offer a non-antibiotic alternative to macrolides, but their place in therapy needs clearer definition through head-to-head or add-on trials.

Non-Antibiotic Macrolide Analogs: EP395 is an oral macrolide without antibacterial activity. In a Phase 2a trial (n = 61), the drug was well tolerated and reduced inflammatory markers, though clinical efficacy remains unproven [99]. These agents could retain immunomodulatory benefits of macrolides while avoiding resistance.

### 5.6. Present and Future of Macrolides in COPD

Are macrolides approaching their therapeutic dusk in COPD, or are we seeing a selective new dawn? GOLD document suggests azithromycin for patients with frequent exacerbations despite optimized inhaler therapy, especially in former smokers [34]. Their future depends on patient phenotype, emerging alternatives, and practical considerations.

Efficacy and Selection: While roflumilast and ensifentrine offer modest or uncertain benefits, and biologics like dupilumab show strong efficacy in eosinophilic COPD, macrolides remain efficacious across a broader population, including non-eosinophilic patients. Trials for non-T2 biologics are ongoing, but current data are preliminary and not yet comparable to approved or soon-to-be-approved T2 biologics. Should non-T2 biologics prove efficacious, they could challenge the role of azithromycin in the treatment of COPD patients with frequent exacerbations.

Safety and Access: Biologics avoid antibiotic resistance but may face cost and accessibility barriers. Macrolides carry risks of resistance, GI intolerance, and ototoxicity, but long-term data are largely reassuring [91,93,94,95]. Their low cost, oral route, and availability make them a pragmatic option, especially where newer therapies are inaccessible or unsuitable.

In conclusion, macrolides in COPD are not in full sunset nor in complete dawn—they occupy a transitional space. Their role is shifting from broad to more selective use, particularly in patients prone to infection or with neutrophilic COPD. If antibiotic resistance increases or new macrolide analogs succeed, their use may evolve further. Thus, their immunomodulatory effects—independent of antimicrobial action—may persist into a therapeutic sunrise. For now, macrolides remain a core, if reshaped, component of the COPD treatment armamentarium.

The main clinical scenarios in which long-term macrolide therapy may be considered in COPD are summarized in Table 5. A comparative overview of macrolides and other current or emerging maintenance therapies for COPD is presented in Table 6.

## 6. Macrolides in Transplantation and Beyond: A Role in Rare Lung Diseases

### 6.1. Long-Term Macrolides in the Treatment of Chronic Allograft Lung Dysfunction (CLAD)

Lung transplantation (LTx) is a recognized treatment for end-stage lung diseases, but its long-term success is often undermined by CLAD—a progressive decline in lung function that affects nearly 50% of patients by 5 years and is associated with high mortality [102,103,104]. CLAD encompasses various phenotypes and is diagnosed when a progressive FEV_1_ decline (>20% from the best postoperative value) is observed in the absence of alternative causes [103].

Azithromycin was initially tested in CLAD following encouraging results in cystic fibrosis. Early open-label studies (azithromycin 250 mg three times weekly or alternate days) reported improved or stabilized FEV_1_ in 43–83% of patients [105,106,107,108]. Subsequent larger studies confirmed these results, showing functional improvement in 30% and stabilization in 46% of patients after 3 months of treatment [109], with benefits more evident in early-stage (possible) CLAD [110]. Clarithromycin was also investigated; it showed a beneficial effect on FEV_1_ in around 40% of patients [111], though with drug interaction concerns (e.g., cyclosporine).

An RCT of azithromycin in 46 patients found no difference in FEV_1_ in the intention-to-treat analysis, but the open-label crossover arm suggested significant functional improvement (306 mL) [112]. Long-term follow-up (up to 5 years) confirmed FEV_1_ stabilization in previously untreated patients, though survival and progression were unaffected [113]. A meta-analysis reinforced the short-term efficacy of azithromycin but highlighted the lack of long-term data [114].

These findings contributed to the definition of azithromycin-responsive allograft dysfunction (ARAD), a CLAD phenotype in which patients recover >90% of best post-transplant FEV_1_ [115]. Current guidelines recommend an 8-week trial of azithromycin in suspected cases [103].

Azithromycin has also been studied as prophylaxis. A two-year RCT starting post-discharge showed improved CLAD-free survival and better lung function, although not overall survival [116]. These findings were extended in a 7-year follow-up [117]. Conversely, a perioperative trial found no functional benefit, though BAL neutrophilia and IL-8 levels decreased [118]. Retrospective studies yielded mixed results, with one showing reduced CLAD incidence with early azithromycin [119], and another showing no effect [120].

The role of azithromycin in modulating the lung microbiome is unclear. A post-hoc RCT analysis found no impact on microbial burden or diversity [121], though response to azithromycin correlated with higher baseline microbial load in another study [122].

Summarising, azithromycin may be efficacious in a proportion of patients with CLAD by stabilising FEV_1_, and may have a role as prophylactic treatment. When a possible CLAD is diagnosed, a trial of azithromycin may reveal the ARAD phenotype. Literature and guidelines are not conclusive on the best course between the two options. This has been confirmed in an international survey on azithromycin use in different LTx centres. Nearly one-third of the centres used azithromycin prophylactically, while the majority of the respondents used the drug in the presence of a decline in FEV_1_ > 10% without an established CLAD. Even intra-centre variability exists [123].

### 6.2. Long-Term Macrolides in the Treatment of Bronchiolitis Obliterans Syndrome (BOS) After Haematopoietic Stem Cell Transplant (HSCT)

BOS is a rare but serious complication of the allogenic HSCT, characterized by a progressive obstructive ventilatory defect, often associated with a mosaic pattern on CT. The current definition of BOS after HSCT includes a FEV_1_ < 75%, with a decline >10% from the baseline before HSCT, unresponsiveness to bronchodilators, and a functional obstruction (FEV_1_/FVC < 70%) [124,125].

Following the results in CLAD, azithromycin was tested in BOS. An initial open-label study in 8 patients showed improvements in FEV_1_ and FVC [126], but a subsequent RCT in 24 patients found no benefit on lung function or symptoms after 12 weeks [127].

Azithromycin has also been explored as BOS prophylaxis. A large retrospective study (with about 1200 subjects) showed no preventive benefit and even worse outcomes in the treated group [128]. More concerning, the ALLOZITHRO RCT was terminated early due to a higher rate of relapse of hematologic disease in the azithromycin group, with no observed advantage in BOS prevention [129]. A meta-analysis confirmed the absence of benefit in FEV_1_ decline at 12 or 24 weeks [130].

Based on the findings of the ALLOZITHRO trial, fearing an increased rate of haematologic relapses, two retrospective studies addressed this issue. The first evaluated the use of azithromycin for any reason and any duration in patients already diagnosed with BOS. The authors found that patients receiving azithromycin did not present an increased risk for haematological relapses. However, there was an increased risk for secondary neoplasms in patients exposed to the drug [131]. The second study compared patients using azithromycin for any reason, irrespective of the diagnosis of BOS. In this study, an increased prevalence of haematological relapses following azithromycin use were seen in a subgroup of patients treated with anti-thymoglobulin due to a matched unrelated donor [132].

An explanation for these different complications may be the timing of the macrolide administration. In the early post-transplant phase, such as in a prophylactic use, azithromycin may reduce the efficacy of the HSCT against the haematological disease, leading to disease relapse. However, after BOS development, which typically occurs 3 to 12 months after the transplant, the immunomodulant effect of the drug may be deleterious in the immune response against other neoplasms [131].

Macrolides, in particular azithromycin, have also been studied in combination with inhaled corticosteroids (fluticasone) and anti-leukotriene agents (montelukast) (FAM). An open-label study of 32 participants found that only 6% of patients declined by more than 10% in FEV_1_, with an improvement in patient reported outcomes and 6-min walking test distance [133]. Based on this data, FAM is suggested by the international guidelines for BOS after HSCT [134]. However, a retrospective case–control study of the patients included compared to a historical cohort found no difference in the rate of decline of FEV_1_ in the treated patients [135]. Therefore, larger RCTs are warranted to clarify the role of FAM in BOS after HSCT, especially in the light of the risk of neoplasms and haematological relapses associated to azithromycin.

### 6.3. Long-Term Macrolides in the Treatment of Diffuse Lung Parenchymal Diseases (DPLDs)

Diffuse parenchymal lung diseases (DPLDs) encompass a broad spectrum of conditions that typically affect the interstitium or the alveolar space of the lung. DPLDs typically present an initial prolonged inflammatory injury that may be followed by the development of interstitial fibrosis [136]. Therefore, macrolides have been studied in preclinical DPLD models to evaluate their known anti-inflammatory activity and potential antifibrotic properties. Azithromycin showed efficacy in reducing collagen and fibronectin production and myofibroblast differentiation in cultures of both control and pathology-derived lung fibroblasts [137]. In the bleomycin-treated mice, the most common model of experimentally induced lung fibrosis, 14-ringed macrolides such as roxithromycin and clarithromycin and the 12-ringed EM703, a derivative of erythromycin, showed an antifibrotic effect [138,139,140]. No study has investigated this class of antibiotics as antifibrotics in a clinical trial.

However, owing to their pleiotropic effects, macrolides have been evaluated in several DPLDs.

In idiopathic pulmonary fibrosis (IPF), the most common and severe idiopathic DPLD, azithromycin has been studied for infection prevention and IPF-related cough. A retrospective cohort study reported reduced non-elective hospitalizations and antibiotic use in patients with frequent infections treated with azithromycin (250 mg three times weekly for one year) [141]. In a placebo-controlled trial, azithromycin (500 mg three times weekly for 12 weeks) did not improve cough frequency or severity [142]. Post-hoc analysis revealed reduced sputum bacterial diversity and emergence of resistant *Streptococci* in treated patients [9].

In sarcoidosis, azithromycin has been studied both in combination antimycobacterial regimens and as monotherapy for chronic cough. The CLEAR protocol (Combination of Levofloxacin, Ethambutol, Azithromycin, Rifampicin) was based on the hypothesis that mycobacteria play a pathogenetic role in sarcoidosis. Initial pilot studies showed improved lung function and reduced cutaneous lesions [143,144], but an RCT in pulmonary sarcoidosis found no functional benefit and even reduced quality of life in the active arm [145]. High discontinuation rates due to pill burden and/or side effects were also reported.

Long-term azithromycin alone has been tested in an open-label trial in 21 sarcoidosis patients with chronic cough. Improvements were observed in objective cough counts and quality-of-life scores, with good tolerability [146]. Interestingly, azithromycin seemed not to change the inflammatory burden of sarcoidosis, with only a slight but significant reduction in neutrophils [147].

Long-term macrolide therapy has been studied as a treatment for Cryptogenic Organising Pneumonia (COP), an inflammatory DPLD characterised by bilateral ground glass opacities and consolidation that may progress to fibrosis. This disease, normally treated with long courses of steroids, tends to relapse as steroids are discontinued. In patients with histologically confirmed COP and no signs of respiratory failure, clarithromycin 500 mg twice a day for three months was not inferior to prednisolone 0.5–1 mg/kg. A retrospective study of 62 patients showed a similarly high response rate in both regimens, with fewer relapses and side effects in the clarithromycin group [148].

However, long-term macrolide therapy has its pinnacle, as standard of care, in a rare DPLD: diffuse panbronchiolitis (DPB). This disease, typically diagnosed in Japan, is characterised by a persistent respiratory inflammation with bilateral micronodular lesions on chest CT. The natural history of DPB has changed since 1985, when long-term low-dose erythromycin showed efficacy in treating the disease [149]. To avoid side effects and pill burden of erythromycin, clarithromycin (200 mg QD) and azithromycin (500 mg QD for 3 months, then 500 mg three times weekly) had been studied with good results in improving and then stabilising clinical, functional and radiological abnormalities [150,151]. However, in DPB, macrolide treatment needs a long duration (up to 4 years) to avoid disease relapse, with unknown effects on antibiotic resistance.

In conclusion, in these diverse and often underexplored respiratory conditions—from CLAD and BOS to different DPLDs—macrolides have shown both flickers of promise and shadows of uncertainty. Their role remains variably defined, balancing potential benefits against emerging risks. Whether this represents a therapeutic dusk or the glimmer of a new dawn will depend on future trials, refined patient selection, and our evolving understanding of the role of inflammation across the spectrum of respiratory diseases.

## 7. Community and Environmental Risks Associated with Long-Term Use of Macrolides

While long-term macrolide therapy can be beneficial in selected patients with chronic respiratory diseases, its use poses important risks for the wider community and the environment. Prolonged exposure may select for macrolide-resistant organisms in the commensal flora, with potential for transmission to other individuals. Randomized controlled trials, such as BAT and BLESS, documented marked increases in macrolide-resistant *Streptococcus pneumoniae* and other *Streptococci* after 6–12 months of treatment, with resistance rates rising to as high as 88% in treated cohorts [15,20]. Such resistance may persist for months after treatment cessation and can undermine the efficacy of macrolides for common community-acquired infections [152,153]. This effect may increase the known antimicrobial resistance of macrolides that is already rising worldwide [154].

At the environmental level, macrolides are frequently detected in wastewater and surface waters due to incomplete removal during conventional treatment processes [155]. Recent wastewater-based epidemiology in Spain detected high levels of azithromycin in sewage, along with a broad spectrum of antimicrobial resistance genes—including integron-associated intl1—underscoring the role of wastewater as a reservoir and dissemination pathway for resistance determinants [156]. Ecotoxicological assessments identify clarithromycin and azithromycin as high-risk compounds for environmental selection of resistance and as potential hazards for aquatic organisms [157].

Long-term macrolide exposure in agricultural soils can also reshape microbial communities and their ‘resistomes’ (the collection of all antimicrobial resistance genes present in a microbiota) and their ‘mobilomes’ (the ensemble of mobile genetic elements capable of transferring these genes). Experimental field studies have shown that repeated high-dose applications (10 mg/kg) significantly increase the abundance of mobile genetic elements and macrolide resistance genes, while even decade-scale exposure to lower doses (0.1 mg/kg) can alter resistome and mobilome composition, enriching resistance gene diversity and horizontal transfer potential [158]. Such environmental reservoirs may contribute to the persistence and spread of antimicrobial resistance far beyond the treated patient population. These direct and indirect effects highlight the need for targeted and judicious use of macrolides, reserving them for carefully selected patients with the chronic respiratory diseases discussed in the previous chapters, in whom the clinical benefits clearly outweigh potential risks not only for the individual patient, but also for the community and the environment.

## 8. Limitations

This review has limitations. Firstly, it is a narrative review, and therefore no formal systematic search or quantitative synthesis was performed; the selection of studies may be influenced by publication bias and the authors’ judgment. Secondly, the study heterogeneity across trials with regard to design, populations, and outcomes limits the comparability of results. Finally, while we attempted to provide a comparative framework with alternative therapies, the indirect comparisons across different drug classes and diseases should be interpreted with caution.

## 9. Conclusions

In this narrative review, we revisited the evidence supporting long-term macrolide therapy across major chronic respiratory diseases, placing their current role into context with alternative and emerging therapeutic options. As summarized in Figure 2, the relevance of macrolides depends on specific patient phenotypes within each condition, emphasizing the importance of an individualized approach to therapy. Overall, with regard to macrolides, chronic respiratory diseases show a landscape in transition: in some settings long-term macrolides are moving toward dusk, while in others they may still rise to a new dawn, maintaining a place in everyday clinical practice.

## Figures and Tables

**Figure 1 antibiotics-14-01061-f001:**
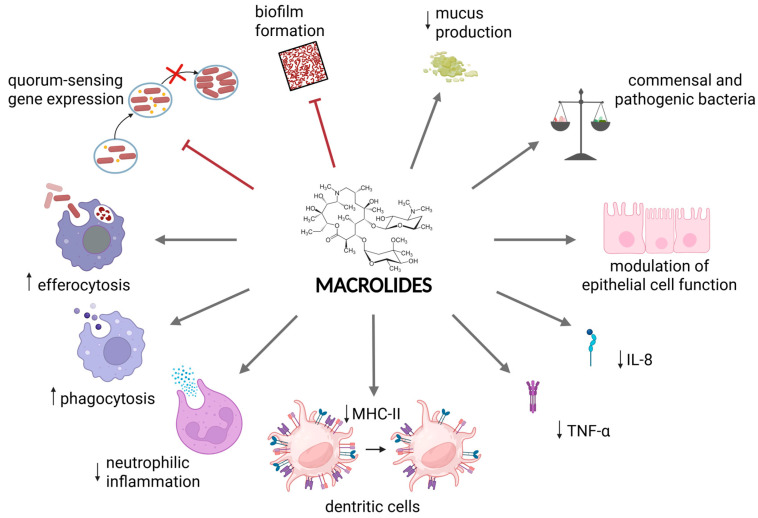
Overview of the multifaceted actions of macrolides in chronic respiratory diseases. Macrolides reduce bacterial virulence by inhibiting quorum sensing and biofilm formation [10,13]. They also support the host’s initial immune response by decreasing both the volume and viscosity of sputum [13,14,15,16,17,18]. Furthermore, macrolides can modulate the composition of the airway and gut microbiota, potentially restoring the balance between commensal and pathogenic bacteria [7,8], and can modulate epithelial cell function [19]. Their anti-inflammatory effects include suppression of pro-inflammatory cytokine and chemokine release (e.g., IL-8, TNF-α), thereby limiting the recruitment of additional immune cells to the lungs [3]. Macrolides also diminish T-cell activation and proliferation, partly through downregulation of co-stimulatory molecules on antigen-presenting cells and, in the case of azithromycin, reduce expression of major histocompatibility complex class II on dendritic cells and neutrophilic inflammation [4]. Conversely, macrolides may enhance certain immune functions by promoting phagocytosis, efferocytosis, and microbial clearance [5]. The effects of macrolides are represented as inhibition (red arrow) and implication (grey arrow). Image created with BioRender.com. (accessed on 1 August 2025).

**Figure 2 antibiotics-14-01061-f002:**
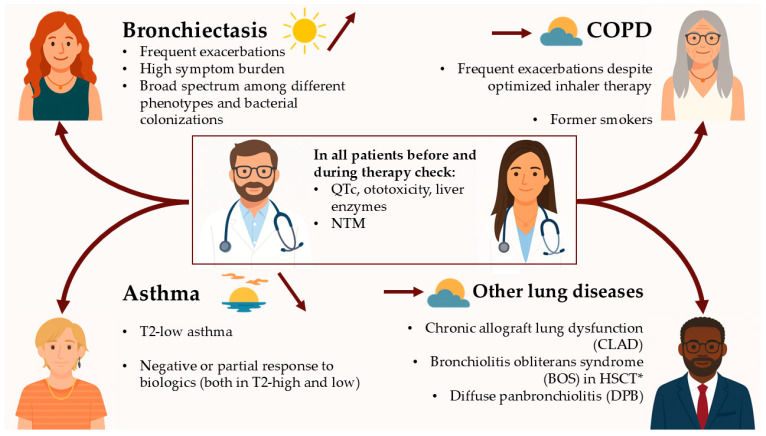
Potential indications for long-term macrolide therapy across chronic respiratory diseases. The sun icons represent the therapeutic trajectory: a high, bright sun (↗) indicates consolidated or growing evidence and clinical use; a mid-level sun (→) represents benefit in selected settings; and a low sun at sunset (↘) reflects a declining or more limited role. Abbreviations: HSCT, Hematopoietic Stem Cell Transplantation; NTM, Nontuberculous Mycobacteria; QTc, Corrected QT interval; T2, type-2. * In the context of FAM therapy (fluticasone, azithromycin, montelukast).

**Table 1 antibiotics-14-01061-t001:** Proposal for Selecting Patients for Long-Term Macrolide Therapy in Bronchiectasis.

Clinical Scenario	Rationale
≥3 exacerbations/year without (and with) *P. aeruginosa* colonization	Macrolides reduce exacerbation rates with higher efficacy in frequent exacerbators [14,15,20].
1–2 exacerbations/year or early-stage disease	Significant benefit even in patients with fewer exacerbations [23,30].
High symptom burden (e.g., daily cough, sputum production, low QoL-B-RSS)	Macrolides improve sputum purulence, reduce daily sputum volume, and patients with high symptom burden but few prior exacerbations derive similar benefits in reducing exacerbations as frequent exacerbators [13,15,16,17,18,20,21].

**Table 2 antibiotics-14-01061-t002:** Comparison of macrolides with other present and future maintenance therapies in bronchiectasis. The range reflects the reduction in exacerbations (per year) in patients treated with corresponding drug compared with placebo, as reported in the aforementioned trials.

Therapy	Description	Relative Reduction in Exacerbations
Macrolides	Oral azithromycin (dosing ranging from 250 mg daily, 250 or 500 mg three times weekly), and erythromycin (dose ranging from 250 to 400 mg twice daily) [16,28]; anti-inflammatory and antimicrobial activity.	43–78% [14,15,20,21]
Inhaled antibiotics (colistin, gentamycin, tobramycin)	Indicated in *Pseudomonas aeruginosa* colonization; local antibacterial effect; requires nebulization device.	22–25% [29]
Brensocatib	Oral DPP-1 inhibitor; targets neutrophilic inflammation; no antimicrobial effect; approved for frequent exacerbators.	20–21% [32]
Inhaled corticosteroids	Consider only in eosinophilic inflammation or asthma overlap.	30% * [31]

* Percentage refers to patients with eosinophilic bronchiectasis (blood eosinophils ≥ 300 cells/μL) [31].

**Table 3 antibiotics-14-01061-t003:** Proposal for Selecting Patients for Long-Term Macrolide Therapy in Asthma.

Clinical Scenario	Rationale
T2-low asthma phenotype	Greater benefit observed in non-eosinophilic asthma [37,44].
Negative or partial response to biologics (both in T2-high and T2-low phenotype)	Azithromycin may provide added control even with biologics ongoing [61].
Access or cost constraints limiting biologic use	Azithromycin is a low-cost and widely available option.

**Table 4 antibiotics-14-01061-t004:** Comparison of macrolides with biologic therapies available in asthma. The range reflects the reduction in exacerbations (per year) in patients treated with corresponding drug compared with placebo, as reported in the aforementioned trials.

Therapy	Description	Relative Reduction in Exacerbations
Macrolides	Oral azithromycin three times weekly for ≥ 6 months [35]; broad efficacy among different phenotypes, possibly higher among non-eosinophilic phenotype.	0–41% [37,39,40]
Omalizumab	Humanized monoclonal anti-IgE antibody; indicated for allergic asthma [58].	23–60% [58]
Mepolizumab	Humanized monoclonal anti-IL-5 antibody; indicated for eosinophilic asthma [58].	34–62% [58]
Benralizumab	Humanized monoclonal anti-IL-5Rα antibody; indicated for eosinophilic asthma [58].	28–61% [58]
Dupilumab	Humanized monoclonal anti-IL-4Rα antibody; indicated for T2-high asthma [58].	41–68% [58]
Tezepelumab	Humanized monoclonal anti-TSLP antibody; indicated for both T2-high and T2-low asthma [64].	47–63% [64]

**Table 5 antibiotics-14-01061-t005:** Proposal for Selecting Patients for Long-Term Macrolide Therapy in COPD.

Clinical Scenario	Rationale
Frequent exacerbations despite optimized inhaler therapy (triple therapy or LABA/LAMA if ICS contraindicated)	Macrolides reduce exacerbation frequency by 25–50% (in selected patients and with a tendency to diminish over time), comparable to or greater than other preventive strategies [65,66,69,71,75,76,78].
Former smoker	Efficacy is reduced in current smokers [70].
Neutrophilic inflammation or frequent bacterial infections/colonization	Macrolides offer both immunomodulatory and antimicrobial effects, efficacious in patients prone to infection or neutrophilic phenotypes [100,101]. Conflicting data in eosinophilic phenotype [74,77].

**Table 6 antibiotics-14-01061-t006:** Comparison of macrolides with other present and future maintenance therapies in COPD. The range reflects the reduction in exacerbations (per year) in patients treated with corresponding drug compared with placebo, as reported in the aforementioned trials.

Therapy	Description	Relative Reduction in Exacerbations
Macrolides	Oral azithromycin (250 mg daily or 500 mg three times weekly) or erythromycin (250 mg twice daily) [34]; broad efficacy among different phenotypes.	~25–50% in selected patients [65,66,69,71,75,76,78]
Roflumilast	Oral PDE4 inhibitor; indicated for CB phenotype and frequent exacerbations.	14–26% (higher efficacy in CB) [83,84]
Dupilumab	Biologic targeting IL-4Rα; for COPD with blood eos ≥ 300 cells/µL, CB and frequent exacerbations despite triple therapy [34].	30–34% [94,95]
Mepolizumab	Biologic targeting IL-5; for COPD with blood eos ≥ 300 cells/µL and frequent exacerbations despite triple therapy.	21% [93]
Ensifentrine	Inhaled dual PDE3/4 inhibitor; bronchodilator with anti-inflammatory effect.	~40% (context-dependent) [96,97]

## Data Availability

The data presented in this study are available within the article.

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
