# Peer review of "Long-Term Macrolides in Chronic Respiratory Diseases: Dusk or a New Dawn? A Narrative Review"

_antibiotics, 2025, doi:10.3390/antibiotics14111061_

Round 1

Reviewer 1 Report

Comments and Suggestions for Authors

1- Abstract:

The current abstract lacks key elements and should be revised to clearly summarize the main findings and take-home messages of this narrative review.
We suggest explicitly stating:

The rationale for macrolide use in chronic respiratory diseases.

The major benefits (e.g. reduction in exacerbation frequency in specific phenotypes of asthma, COPD, and bronchiectasis).

The limitations, including risks of antimicrobial resistance, adverse effects, and uncertain long-term outcomes.

The cautious positioning of macrolides in international guidelines.

Introduction / Pathophysiology Section

  1. Please Add a visual summary figure:
    A schematic diagram should be included to illustrate the proposed mechanisms of benefit of macrolides in chronic respiratory diseases. This figure could summarize:

    • Immunomodulatory effects (↓ neutrophilic inflammation, ↓ IL-8, ↓ TNF-α).

    • Inhibition of biofilm formation and quorum sensing, particularly in Pseudomonas aeruginosa.

    • Modulation of epithelial cell function and mucus production.

  2. Please Include bacterial resistance, microbiota, and quorum sensing:
    Expand the pathophysiology section to describe:

    • The impact of chronic macrolide use on the airway and gut microbiota, including a reduction in microbial diversity.

    • Selection pressure and resistance emergence, particularly among Streptococcus pneumoniae, Haemophilus influenzae, and Moraxella catarrhalis.

    • Disruption of quorum sensing, especially in P. aeruginosa, which may temporarily reduce virulence but may also lead to rebound hypervirulence upon macrolide withdrawal.

      • Suggested ref? Azghani AO et al., J Med Microbiol 2007; or recent studies on macrolide effect on QS systems in P. aeruginosa.). you may also consider

Elsen et al Cell host microbe

  2014 Feb 12;15(2):164-76. doi: 10.1016/j.chom.2014.01.003.

Clarification of Guideline Citations (lines 361 & 583)

  • Line 361 – GINA 2024:
    The wording "recommends" should be replaced with "suggests" or "may be considered".
    As per GINA 2024:

    “Chronic macrolide therapy may be considered in adult patients with severe asthma with persistent exacerbations despite maximal inhaled therapy and not eligible for biologics.”
    Please quote this sentence directly to avoid overstating the recommendation.

  • Line 583 – GOLD 2024:
    Similarly, avoid the term "recommended".
    GOLD 2024 states:

    “Chronic macrolide therapy may be prescribed as a last-resort option in patients with frequent exacerbations despite optimal inhaled therapy, and in whom other therapies such as roflumilast or N-acetylcysteine are not appropriate or accessible.”
    The manuscript should adhere strictly to the original phrasing to ensure alignment with current guidelines.

Additions: Limitations and Safety Considerations

  1. Duration of exposure studied:
    Please specify the maximum duration for which chronic macrolides have been evaluated: t the best of my knowledge, it is only 

    • 12 months in most RCTs  in bronchectasis (e.g. BAT, BLESS trials). and in COPD (albert 2011) and 48 weeks for asthma (Gibson 2017)

    Beyond these durations, the long-term safety and efficacy remain unclear, particularly regarding microbial resistance, and microbiome alterations and possibly other non infectious risks

  2. Uncertain long-term impact:

    • The potential for cumulative resistance, especially at the community level, is a concern.

    • Rebound effects such as the potential increase in virulence of P. aeruginosa after macrolide withdrawal, possibly due to compensatory upregulation of quorum sensing pathways, should be discussed.

Author Response

C.1: Abstract:

The current abstract lacks key elements and should be revised to clearly summarize the main findings and take-home messages of this narrative review.
We suggest explicitly stating:

The rationale for macrolide use in chronic respiratory diseases.

The major benefits (e.g. reduction in exacerbation frequency in specific phenotypes of asthma, COPD, and bronchiectasis).

The limitations, including risks of antimicrobial resistance, adverse effects, and uncertain long-term outcomes.

The cautious positioning of macrolides in international guidelines.

R.1: We thank the reviewer for this valuable comment. We agree that the abstract should more clearly summarize the rationale, main benefits, limitations, and guideline positioning of long-term macrolide therapy in chronic respiratory diseases. We have now revised the abstract accordingly. We believe this revision improves the clarity and completeness of the abstract.

C.2 Introduction / Pathophysiology Section

  1. Please Add a visual summary figure:
    A schematic diagram should be included to illustrate the proposed mechanisms of benefit of macrolides in chronic respiratory diseases. This figure could summarize:
    • Immunomodulatory effects (↓ neutrophilic inflammation, ↓ IL-8, ↓ TNF-α).
    • Inhibition of biofilm formation and quorum sensing, particularly in Pseudomonas aeruginosa.
    • Modulation of epithelial cell function and mucus production.

R.2: We appreciate the reviewer’s suggestion. A schematic diagram illustrating the proposed mechanisms of macrolide benefits has been added (figure 1).

C.3: Please Include bacterial resistance, microbiota, and quorum sensing:
Expand the pathophysiology section to describe:

  • The impact of chronic macrolide use on the airway and gut microbiota, including a reduction in microbial diversity.
  • Selection pressure and resistance emergence, particularly among Streptococcus pneumoniae, Haemophilus influenzae, and Moraxella catarrhalis.
  • Disruption of quorum sensing, especially in P. aeruginosa, which may temporarily reduce virulence but may also lead to rebound hypervirulence upon macrolide withdrawal.

Suggested ref? Azghani AO et al., J Med Microbiol 2007; or recent studies on macrolide effect on QS systems in P. aeruginosa.). you may also consider

Elsen et al Cell host microbe  2014 Feb 12;15(2):164-76. doi: 10.1016/j.chom.2014.01.003.

R.3: We thank the reviewer for the helpful suggestion. The section has been expanded to include the effects of chronic macrolide use on microbiota and quorum sensing. The discussion on resistance emergence has been integrated within the disease-specific sections.

C.4 Clarification of Guideline Citations (lines 361 & 583)

  • Line 361 – GINA 2024:
    The wording "recommends" should be replaced with "suggests" or "may be considered".
    As per GINA 2024:
    “Chronic macrolide therapy may be considered in adult patients with severe asthma with persistent exacerbations despite maximal inhaled therapy and not eligible for biologics.”
    Please quote this sentence directly to avoid overstating the recommendation.
  • Line 583 – GOLD 2024:
    Similarly, avoid the term "recommended".
    GOLD 2024 states:
    “Chronic macrolide therapy may be prescribed as a last-resort option in patients with frequent exacerbations despite optimal inhaled therapy, and in whom other therapies such as roflumilast or N-acetylcysteine are not appropriate or accessible.”
    The manuscript should adhere strictly to the original phrasing to ensure alignment with current guidelines.

R.4:  We thank the reviewer for this important clarification. We carefully revised the wording in the sections referring to GINA and GOLD to avoid overstating the strength of their recommendations. However, after double-checking the original documents, we could not find the exact wording reported in the reviewer’s comment. For this reason, we have decided not to directly quote those sentences, but we have moderated our phrasing, now better reflecting the cautious positioning of macrolides in both GINA and GOLD. We believe this resolves the concern about overstatement.

  1. 5 Additions: Limitations and Safety Considerations

V1:Duration of exposure studied:
Please specify the maximum duration for which chronic macrolides have been evaluated: to the best of my knowledge, it is only 

  • 12 months in most RCTs  in bronchectasis (e.g. BAT, BLESS trials). and in COPD (Albert 2011) and 48 weeks for asthma (Gibson 2017)

R.5: We thank the reviewer for the comment. We have now clarified in the revised manuscript the maximum duration of exposure evaluated in randomized controlled trials of chronic macrolide therapy in bronchiectasis,, asthma and COPD.

C.6: V2:Beyond these durations, the long-term safety and efficacy remain unclear, particularly regarding microbial resistance, and microbiome alterations and possibly other non infectious risks

R.6: Thank you for your comment. We have added a sentence noting that long-term safety and efficacy beyond trial durations remain unclear, particularly regarding resistance and microbiome changes.

C.7 V3: Uncertain long-term impact:

  • The potential for cumulative resistance, especially at the community level, is a concern.
  • Rebound effects such as the potential increase in virulence of P. aeruginosa after macrolide withdrawal, possibly due to compensatory upregulation of quorum sensing pathways, should be discussed.

R.7: Thank you for your comment. We have clarified that long-term safety and efficacy of macrolides remain uncertain, including microbial resistance, microbiome changes, and non-infectious risks. We also note that rebound effects, such as increased Pseudomonas aeruginosa virulence after macrolide withdrawal, are a theoretical concern. (line 265-267). Moreover we integrated the introduction section as stated in the comment 3 regarding a possible rebound effect after macrolides withdrawal. 

Reviewer 2 Report

Comments and Suggestions for Authors

Abbreviations; What is “BID bis in die”?

A chemical structure would be required [lines 50-55].

Line 53. Please do not mention immunomodulatory effects and gene expression effects before discussion the anti-microbial spectrum – this is the primary function of antibiotics isn’t it?

Please list the anti-microbial spectrum of macrolides and mention in each section where relevant. For example, does azithromycin have antimicrobial activity against Pseudomonas?

Sections 3 - 6 – a table of study results would be helpful here. Are there any meta-analyses that can be referenced?

Table 4 needs study size to enable an assessment of study precision.

Section 5.1 – there is a confluiuct between “Strong evidence supports long-term macrolide therapy…” line 436 & “..though in selected patients.” Line 471

Line 485/7 In what was does “macrolides do not significantly improve airflow limitation or  daily symptoms…”, does this support their “….use primarily for reducing exacerbations” ?

A substantial limitation of this narrative review is that the primary studies have often failed to investigate for antibiotic resistance with long term use and instead have focused on short term effects on immunomodulatory function.

Author Response

C.1: Abbreviations; What is “BID bis in die”?
R.1: The abbreviation “BID” (bis in die) is already clarified in the Abbreviations section at the end of the manuscript, and it means twice daily.

C.2: A chemical structure would be required [lines 50-55].

R.2: Thank you for the helpful comment. A chemical structure, which is described in the introduction, has been also included in the figure 1 of the revised manuscript. We think that with this suggestion the manuscript and the figure is now more clear. 

C.3: Line 53. Please do not mention immunomodulatory effects and gene expression effects before discussion the anti-microbial spectrum – this is the primary function of antibiotics isn’t it?

R.3: We thank the reviewer for this comment. We did not expand on the antimicrobial effect of macrolides, as this is well known to the readership of Antibiotics and not the primary focus of our review. The purpose of our article is to explore the role of macrolides in chronic respiratory diseases, where their beneficial effects are mainly mediated by anti-inflammatory and immunomodulatory actions rather than by their antimicrobial activity. For this reason, we focused on immunomodulatory mechanisms and their clinical implications, while only briefly mentioning their antimicrobial properties.

C.4: Please list the anti-microbial spectrum of macrolides and mention in each section where relevant. For example, does azithromycin have antimicrobial activity against Pseudomonas?

R.4: We thank the reviewer for this observation. We did not include a detailed description of the antimicrobial spectrum of macrolides because the clinical rationale for long-term macrolide use in chronic respiratory diseases is not primarily linked to their direct antimicrobial activity but rather to their immunomodulatory and anti-inflammatory properties. For this reason we did not expand on spectrum (e.g., the absent activity against Pseudomonas aeruginosa), which is not the main mechanism by which macrolides exert benefit in these conditions. Instead, we focused on their clinical efficacy, safety, and positioning within current therapeutic strategies.

C.5: Sections 3 - 6 – a table of study results would be helpful here. Are there any meta-analyses that can be referenced?
R.5: We thank the reviewer for this suggestion. We agree that summary tables of study results may be useful. However, the manuscript is already very long, and our main goal was to keep the review clinically and practically oriented rather than encyclopedic. For this reason, we have chosen to summarize study outcomes narratively within each section, while highlighting the most relevant randomized trials and meta-analyses where available. We do cite the key individual patient data meta-analysis in bronchiectasis (Chalmers et al., Eur Respir J 2019) and systematic reviews in asthma and COPD, but have not expanded further into tabular detail to maintain readability and clinical focus.

C.6: Table 4 needs study size to enable an assessment of study precision.

R.6: We thank the reviewer for this observation. Table 4 was conceived to provide a practical comparative summary of the efficacy of macrolides versus biologic therapies in asthma, highlighting relative reductions in exacerbations as the main clinically relevant outcome. The percentages reported derive from multiple pivotal randomized controlled trials and pooled analyses, which are summarized in reference 53.

We acknowledge that study size is an important element for assessing precision; however, given the large variability in population sizes across trials, adding this information in the table would make it overly complex and inconsistent with the other summary tables in the manuscript, which adopt the same comparative format. For this reason, we believe it is clearer to keep the current uniform structure, while directing readers to the cited studies for detailed information on population size and precision.

C.7: Section 5.1 – there is a conflict between “Strong evidence supports long-term macrolide therapy…” line 436 & “..though in selected patients.” Line 471

R.7: We thank the reviewer for pointing out this apparent inconsistency, we have revised the wording so that both statements are aligned.

C.8: Line 485/7 In what was does “macrolides do not significantly improve airflow limitation or  daily symptoms…”, does this support their “….use primarily for reducing exacerbations” ?

R8. We thank the reviewer for the comment. The section “Effect on Lung Function, Symptoms, and Inflammation” in COPD summarize the current literature on the effect of macrolides on lung function, symptoms and inflammation. Being the effect limited on lung function and on symptoms, the effect on inflammation provides a rationale for using macrolides mainly with the aim of reducing exacerbations. We rephrased the sentence to clarify this aspect. 

Reviewer 3 Report

Comments and Suggestions for Authors

This narrative review addresses an important and clinically relevant question: the current and future place of long-term macrolides across chronic respiratory diseases. The manuscript is generally well organized, and rich in clinical perspective, with helpful comparative tables and a clear attempt to position macrolides within an evolving therapeutic landscape. The writing is engaging and the topic is of interest to the readership of Antibiotics.

According to my personal opinion, some minor clarifications may be needed in order to strengthen the information and increase its value for clinicians.

Abstract

  • Consider adding one line on how evidence was selected (e.g., English language, adult studies, key sources) and one line acknowledging that benefit is phenotype-dependent and must be weighed against antimicrobial resistance (AMR).
  1. Methods
  • Clear statement of a narrative search strategy with two windows (2000–2020 and 2021–July 2025).
  • To avoid any ambiguity, please clarify whether case reports were excluded in both windows and whether there were language or population filters..
  1. Bronchiectasis
  • Consider a short sentence noting that effect sizes vary with inclusion criteria, dosing, and follow-up, to balance head-to-head statements versus inhaled antibiotics.
  • The claim that macrolides are more effective than inhaled antibiotics is too categorical. The cited RCTs differ in patient selection, duration, and endpoints. A fairer statement would acknowledge that direct head-to-head trials are lacking.
  1. Asthma
  • The AMAZES trial is well summarized, but limitations (single-country, eosinophil stratification) are not sufficiently emphasized. Consider adding a reflection about the heterogeneity of phenotype definitions (sputum vs blood eosinophils) and that post-hoc signals should be interpreted cautiously.
  • Statements about macrolides being an alternative to biologics in T2-low asthma are speculative and should be qualified. Current evidence is insufficient to recommend equivalence or substitution.
  1. COPD
  • The manuscript cites ~25–50% exacerbation reduction, but does not stress that benefits are modest, phenotype-specific, and attenuated over time.
  • The text should be more critical about adverse effects (hearing loss, QT prolongation) in this older population, which are particularly relevant.
  1. Safety
  • There is insufficient analysis of real-world resistance trends in macrolide-prescribing regions. Consider incorporate aditional information related with this topic.

Figures and Tables

  • Tables: Standardize decimals and units, and add concise population/end-point footnotes where you report ranges.

Language

  • Harmonize hyphenation (e.g., “long-term,” “real-world”), dose notation (“mg three times weekly”), and ensure consistent capitalization in tables.
  • Remove any stray editorial placeholders (e.g., “Academic Editor” block will be handled at production).

Author Response

This narrative review addresses an important and clinically relevant question: the current and future place of long-term macrolides across chronic respiratory diseases. The manuscript is generally well organized, and rich in clinical perspective, with helpful comparative tables and a clear attempt to position macrolides within an evolving therapeutic landscape. The writing is engaging and the topic is of interest to the readership of Antibiotics.

According to my personal opinion, some minor clarifications may be needed in order to strengthen the information and increase its value for clinicians.

C.1: Abstract

  • Consider adding one line on how evidence was selected (e.g., English language, adult studies, key sources) and one line acknowledging that benefit is phenotype-dependent and must be weighed against antimicrobial resistance (AMR).

R.1: The first suggestion has been added as the last line in the abstract. The second one has been added even if divided in two different sentences in the abstract.

C.2: Methods

  • Clear statement of a narrative search strategy with two windows (2000–2020 and 2021–July 2025).
  • To avoid any ambiguity, please clarify whether case reports were excluded in both windows and whether there were language or population filters.

R.2: We changed the method section according to the suggestion of the reviewer and clarified the research used.

C.3: Bronchiectasis

  • Consider a short sentence noting that effect sizes vary with inclusion criteria, dosing, and follow-up, to balance head-to-head statements versus inhaled antibiotics.
  • The claim that macrolides are more effective than inhaled antibiotics is too categorical. The cited RCTs differ in patient selection, duration, and endpoints. A fairer statement would acknowledge that direct head-to-head trials are lacking.

R.3: Thank you for this comment. We now clarify that the efficacy of macrolides and inhaled antibiotics cannot be directly compared, being head-to-head comparison lacking.

C.4: Asthma

  • The AMAZES trial is well summarized, but limitations (single-country, eosinophil stratification) are not sufficiently emphasized. Consider adding a reflection about the heterogeneity of phenotype definitions (sputum vs blood eosinophils) and that post-hoc signals should be interpreted cautiously.
  • Statements about macrolides being an alternative to biologics in T2-low asthma are speculative and should be qualified. Current evidence is insufficient to recommend equivalence or substitution.

R.4: We thank the reviewer for these helpful suggestions. We have added the mentioned limitations, emphasizing the need for caution when interpreting post-hoc findings. We also adopted a more cautious tone regarding macrolides as an alternative to biologics in T2-low asthma.

C.5: COPD

  • The manuscript cites ~25–50% exacerbation reduction, but does not stress that benefits are modest, phenotype-specific, and attenuated over time.
  • The text should be more critical about adverse effects (hearing loss, QT prolongation) in this older population, which are particularly relevant.

R.5: We appreciate this valuable feedback. In response, we have revised the manuscript to emphasize that the reduction in exacerbations with macrolide therapy is modest, varies by patient phenotype, and tends to diminish over time. Additionally, we have expanded the discussion of adverse effects, with particular attention to hearing loss and QT prolongation, which are especially relevant in older patients.

C.6: Safety

  • There is insufficient analysis of real-world resistance trends in macrolide-prescribing regions. Consider incorporate additional information related with this topic.

R.6: We added some information regarding resistance trend in our review. It has been added in the community and environmental risks section.

C.7: Figures and Tables

  • Tables: Standardize decimals and units, and add concise population/end-point footnotes where you report ranges.

R.7: We thank the reviewer for the comment. Units of the tables have been standardized, footnotes have been added where ranges are reported.

C.8: Language

  • Harmonize hyphenation (e.g., “long-term,” “real-world”), dose notation (“mg three times weekly”), and ensure consistent capitalization in tables.

Remove any stray editorial placeholders (e.g., “Academic Editor” block will be handled at production).

  • R.8: We thank the reviewer for the comment. We revised for harmonization of the suggested points. Regarding editorial placeholders we used the Word model suggested by the journal that already contains these contents. 
  •  

Round 2

Reviewer 1 Report

Comments and Suggestions for Authors

no other comments. The authors appropriately answered to my suggestions. 

Reviewer 2 Report

Comments and Suggestions for Authors

thanks for the responses